# Differentiating electrophysiological indices of internal and external performance monitoring: Relationship with perfectionism and locus of control

Alexandra M. Muir[ORCID][1][⦿], Kaylie A. Carbine[1][⦿], Jayden Goodwin[1], Ariana Hedges-Muncy[1], Tanja Endrass[2], Michael J. Larson[1,3]*

1 Department of Psychology, Brigham Young University, Provo, Utah, United States of America,
2 Technische Universität Dresden, Dresden, Germany, 3 Neuroscience Center, Brigham Young University, Provo, Utah, United States of America

⦿ These authors contributed equally to this work.
* michael_larson@byu.edu

**Data Availability Statement:** All data, code used for data analyses, and supplementary materials have been posted to the Open Science Framework

## Abstract

The impact of individual differences on performance monitoring and psychopathology is a question of active debate. Personality traits associated with psychopathology may be related to poor internal performance monitoring (as measured by the error-related negativity [ERN]) but intact external performance monitoring (as measured by the reward positivity [RewP]), suggesting that there are underlying neural differences between internal and external performance monitoring processes. We tested the relationships between individual difference measures of perfectionism, locus of control, and ERN, error-positivity (Pe), and RewP component difference amplitude in a healthy undergraduate sample. A total of 128 participants (69 female, $M(SD)_{age}$ = 20.6(2.0) years) completed two tasks: a modified version of the Eriksen Flanker and a doors gambling task along with the Frost Multidimensional Perfectionism scale, the Rotter Locus of Control scale, and the Levenson Multidimensional Locus of Control scale to quantify perfectionism and locus of control traits, respectively. Linear regressions adjusting for age and gender showed that neither ΔERN nor ΔRewP amplitude were significantly moderated by perfectionism or locus of control scores. Findings suggest that, in psychiatrically-healthy individuals, there is not a strong link between perfectionism, locus of control, and ERN or RewP amplitude. Future research on individual difference measures in people with psychopathology may provide further insight into how these personality traits affect performance monitoring.

## Introduction

Performance monitoring is the ability to assess task execution and make corresponding judgments and alterations to improve results [1]. Age [2], social context [3], personality [4], anxiety levels [5], and working memory span [6] are some of the many factors that play a role in the

(OSF) and can be found at this link: https://osf.io/8pkzu/.

**Funding:** This research was supported by Brigham Young University and partly supported by the German Research Foundation (SFB 940). The funders had no role in study design, data collection and analysis, decision to publish, or preparation of the manuscript.

**Competing interests:** The authors have declared that no competing interests exist.

efficiency of performance monitoring. A growing consensus indicates that personality traits that vary across individuals, such as anxious apprehension, are also consistently associated with increased neural indices of performance monitoring [7]. The role of similar individual difference traits, such as perfectionistic tendencies and locus of control, are less understood.

A method to examine neural reflections of performance monitoring is analysis of event-related potential (ERP) components [8]. The error-related negativity (ERN) is a negative-going deflection in the ERP waveform thought to originate from the anterior cingulate cortex (ACC) that occurs between 50 and 100ms after an incorrect response is made [9–13]. Although there are many theories concerning the functional significance of the ERN, the current consensus is that the ERN represents a monitoring function of cognition or emotional responses associated with performance accuracy and subsequent behavioral adaptation [11,14–17].

Another ERP component where personality trait differences, such as levels of anhedonic depression, are implicated is the reward positivity (RewP; [18]). The RewP is a positive going waveform in response to feedback that occurs approximately 200 to 300 milliseconds after a favorable outcome or positive feedback is presented [17]. When positive feedback is absent (or negative feedback is present), there is a negative deflection in the waveform (previously referred to as the feedback negativity (FN); [17]). Throughout the current paper, many of the manuscripts cited originally investigated the FN. However, due to studies separating the reward-related positivity from the absence of reward that appeared as a negativity, we will refer to this component as the RewP [17]. The RewP increases in amplitude as increasingly positive pictures or rewards are presented to participants [19], changes with the presentation of reward-salient stimuli [20,21], and may serve as a reward prediction error signal indicating the need for future behavior adjustment to obtain desired feedback [20].

Clinical relevance of the ERN and RewP in relation to performance monitoring are seen in multiple studies of individuals with psychopathology. Individuals diagnosed with psychopathology tend to show altered ERN amplitudes when compared to psychiatrically-healthy controls. For instance, there is evidence that people with schizophrenia and autism spectrum disorders (ASD) manifest a smaller ERN amplitude when compared to healthy controls [22–28]. However, recent research indicates that there are non-significant differences in RewP amplitude between those with ASD or schizophrenia and psychiatrically-healthy controls [29–32]. The decrease in ERN amplitude but lack of difference between psychopathology groups in RewP amplitude suggests that individuals with ASD or schizophrenia may have deficits in internal performance monitoring, but not in their ability to use more concrete external feedback to monitor and adjust their performance [30,33], although this finding has not always been consistent in schizophrenia research [34].

Given the findings of differential ERN and RewP amplitude, the ERN and RewP ERP components may be useful in distinguishing if there are specific performance monitoring deficits that occur in individuals with psychopathology. Such a distinction would be significant for treatment aimed to help individuals learn from their mistakes and appropriately adapt their behavior. It would also be beneficial to know what certain aspects of a psychopathology, such as associated character traits, are related to the discrepancies seen between internal and external performance monitoring. As such, we sought to test the relationship between ERN and RewP amplitude in relation to various personality traits in a psychiatrically-healthy sample in order to determine what characteristics might be related to the observed differences between internal and external performance monitoring and may be subsequently useful to focus on in a clinical population.

One personality trait that is often implicated in psychopathology and may affect performance monitoring and related ERP components is perfectionism. Perfectionism includes the pursuit of unrealistic standards of performance and the intolerance of mistakes when trying to

reach said standards [35]. Because expectations of performance are so high, perceived failures are common and are viewed as personal deficiencies [36]. Specifically, in maladaptive perfectionism, individuals tend to set unreachably high-performance expectations and often participate in maladaptive self-criticism, which can be neurotic and harmful to the individual [37–39]. As such, maladaptive perfectionism is a common underlying factor in several psychiatric disorders, including obsessive-compulsive disorder (OCD), obsessive-compulsive personality disorder (OCPD), eating disorders, and anxiety disorders [40–43].

The neural correlates of high perfectionistic tendencies are poorly understood. Recent work suggests perfectionism is associated with increases in performance monitoring—including the processing of errors [44,45]. Traits included in the general term of perfectionism, such as holding extremely high personal standards, fear of negative evaluation, and doubts over actions, are also associated with enhanced ERN amplitudes [45,46]. In addition to these previous findings, people with maladaptive perfectionism tend to larger (i.e., more negative) ERN amplitude relative to individuals with adaptive perfectionism and people without perfectionistic tendencies [44], suggesting that perfectionism plays a role in performance monitoring as indexed by the ERN.

In individuals with anxiety, a disorder with characteristically high levels of maladaptive perfectionism [47], RewP amplitude is blunted which may be indicative of impaired sensitivity to external cues [48]. However, to our knowledge, there are currently no studies that have examined perfectionism and its relationship to the RewP directly, nor in contrast with ERN amplitude in the same sample. Taken together, perfectionism may heighten internal assessment of behavior (as quantified by the ERN) but may dampen or not strongly affect external performance monitoring (as measured by the RewP). The first aim of our study, therefore, was to test the relationship between perfectionistic traits and internal and external indices of performance and reward monitoring as indexed by the ERN and RewP components.

Another personality trait that is often implicated in psychopathology and may be associated with performance monitoring ERP components is locus of control. Locus of control is defined as one's perceived control over his or her environment and situation [49]. Those with a more internal locus of control believe they have greater control over their environment and therefore can influence it, while those with a more external locus of control believe they have little control over their situations and instead the environment influences them. Locus of control and perfectionism are theorized to be related, as those with high perfectionistic standards feel a lack of control over the outcomes of their actions (i.e., they feel they will "never" succeed) much like individuals with an external locus of control [50]. Therefore, it has been suggested that external locus of control moderates the apparent relationship between perfectionistic tendencies and certain psychopathologies, such as post-partum depression [50]. In relation to performance monitoring, internal versus external locus of control may influence how an individual perceives errors because it may change our view as to what or whom is responsible for said errors. For example, an individual with an internal locus of control will attribute outcomes to internal factors, such as skills and abilities. Therefore, errors may be increasingly salient to those who have an internal locus of control when compared to those who attribute errors not to personal reasons, but to external sources, such as the environment. Currently, there are no studies that have tested how internal and external locus of control relates to performance monitoring (ERN) and reward-related (RewP) amplitudes. Thus, the second aim of this study is to test the possible relationship of locus of control as a personality characteristic that is differentially related to the ERN or RewP.

The error positivity or post-error positivity (Pe) is another prominent ERP component that reflects internal performance monitoring. The Pe is a posterior, positive going peak in the ERP waveform that appears approximately 200 to 400 ms after an erroneous response. The Pe is

thought to reflect conscious awareness of error commission [51], as the Pe is much more prominent for conscious errors versus unconscious errors [52]. Pe amplitudes are also positively correlated with perfectionistic characteristics, such as high personal standards or high evaluative concerns, but these findings have not always been consistent [45,53]. Other studies have shown that blunted Pe amplitudes are related to higher levels of perfectionism [54], again suggesting mixed results when examining the Pe and perfectionism. Due to the wide variety of sample sizes in the literature to date (n = 43 [53]; n = 94 [45]; n = 17 [54]) larger-scale studies across a range of perfectionistic tendencies are needed in order to further understand the relationship between perfectionism and the Pe.

Given that personality traits, such as perfectionism and locus of control, may moderate ERN, RewP, and Pe amplitudes, we aimed to study the relationship between perfectionism, locus of control, and these ERP components. For our primary pre-registered analyses, we used difference amplitudes (error minus correct [ERN and Pe] or reward minus loss [RewP]) in order to isolate the specific error- and reward-related activity, rather than using the less-specific ERN or RewP components in isolation. As secondary, exploratory, analyses we used a residualized difference score to account for possible poor reliability associated with subtraction difference scores [55–57]. We first hypothesized that individuals with increased perfectionistic tendencies would exhibit a greater ERN difference amplitude (ΔERN) and a smaller RewP difference amplitude (ΔRewP) compared to those with lower perfectionistic tendencies due to enhanced internal performance monitoring. Second, we hypothesized that those with a more internal locus of control would exhibit larger ΔERN and smaller ΔRewP when compared to those with a more external locus of control due to enhanced internal performance monitoring. Although the primary goal of the present study was to differentiate between the processes of the ERN and RewP, the ΔPe was also examined in an exploratory manner as another neural indicator of internal performance monitoring. We hypothesized a heightened ΔPe would be related to increased perfectionism levels and a more internal locus of control.

## Materials and method

All data, code used for data analyses, and supplementary materials have been posted to the Open Science Framework (OSF) and can be found at this link: https://osf.io/8pkzu/.

### Participants and Procedures

The Institutional Review Board at Brigham Young University approved the current study. Written consent was obtained from each participant prior to participation. The original sample included 181 individuals recruited from undergraduate courses and given course credit for participation. Exclusion criteria were determined *via* self-report and included: being outside the ages of 18 and 55 years, left-handedness, neurological disease, psychiatric disorders, learning disability, or head injury that resulted in loss of consciousness. One participant was excluded from data analysis due to age, and three were excluded due to incomplete questionnaire data. For ΔERN analyses, five additional participants were excluded due to computer malfunction during data collection and 31 participants were excluded for not having enough trials to produce a reliable signal (see Electroencephalogram Recording and Reduction section below). Additionally, 13 participants were excluded for having less than 50% accuracy in the flanker task. The final sample for the ΔERN and ΔPe analyses included 128 individuals (69 female, $M_{age}$ = 20.6 years, $SD_{age}$ = 2.0 years). For the ΔRewP analyses, eight participants were excluded due to computer malfunction during data collection. Eighteen participants were excluded for not completing the doors task, as it was introduced after the initial experiment had begun, and 32 participants were excluded for not having enough trials to produce a

reliable signal (see Electroencephalogram Recording and Reduction). The final sample for the $\Delta$RewP analyses included 119 undergraduates (65 female, $M_{age}$ = 20.5 years, $SD_{age}$ = 2.0 years).

Participants reported for a single laboratory session where written informed consent was first obtained and then a standard demographic questionnaire administered. Subsequently, the following questionnaires were administered in the following order: Beck Depression Inventory-2[nd] edition (BDI-II), Levenson Multidimensional Locus of Control scale (including the Internality subscale (I), Powerful Others subscale (P), and Chance subscale (C)), Frost Multidimensional Perfectionism Scale (F-MPS), Rotter's Locus of Control Scale (Rotter), Penn State Worry Questionnaire (PSWQ), the State-Trait Anxiety Inventory (STAI), and lastly the Obsessive-Compulsive Inventory short version (OCI-R). We list all measures here for sake of completeness and transparency. However, the current study focuses on perfectionism and locus of control and therefore, statistical analyses focused on the measures of perfectionism and locus of control (F-MPS, Rotter, Levenson subscales). Additional questionnaires were used simply as supplementary questionnaires in order to describe our sample. Therefore, no analyses, including correlations with ERP data, were run on BDI-II, OCI-R, and PSWQ. Descriptions of the additional scales can be found in the supplementary materials on OSF (see S1 Appendix on OSF), with means, standard deviations, and ranges for all scales being reported in S1 Table on OSF.

The F-MPS has been used to assess various dimensions of perfectionistic traits and relate perfectionism to various psychiatric disorders [58,59]. The F-MPS includes six subscales, including concern over mistakes (CoM), personal standards (PS), parental expectations (PE), parental criticism (PC), doubts about actions (DaA), and organization (O; [60]). Cronbach's alpha scores for all subscales of the F-MPS tend to be above 0.7 (61). For each question, there are five response choices ranging from strongly disagree (+1) to strongly agree (+5). Scores were summed for each subscale and a total sum was calculated for each participant across all scales (excluding the organization subscale; possible range of scores is 29–145). Per the F-MPS author's recommendation [59], our total score does not include the organizational scale due to the fact that organization is not a major indicator of perfectionism but can be a personality trait found in someone with perfectionistic tendencies. Cronbach's alpha for the F-MPS scale without the organization subscale for our current sample was 0.86 ($M(SD)$ = 82.91(12.01), range = 53–113).

The Rotter scale was used as a measure of locus of control [49]. Twenty-three of the 29 items (6 items were distractors) were scored with a one indicating a more external locus of control and a zero indicating a more internal locus of control. Total scores range from 0–23, with a lower score indicating a more internal locus of control and a high score indicating a more external locus of control. Cronbach's alpha for the Rotter scale in our sample was 0.54 ($M(SD)$ = 9(3.04), range = 3–19).

The Levenson Multidimensional Locus of Control scale was also used in order to quantify internal and external locus of control. Each of the 24 statements included in the scale was rated on a six-point scale and then rescored from -3 to 3 (excluding 0). As there is not a total score for the Levenson scale, the questionnaire was broken down into its three subscales: Internality (Levenson-I), Powerful Others, (Levenson-P), and Chance (Levenson-C). Within the subscales, scores were summed and then a constant of 24 was added to each score in order to get rid of negative values. Each subscale had a minimum score of 0 and a maximum score of 48 points. The overall Cronbach's alpha for the Levenson scale in our current sample was 0.74. When broken down by each subscale, the Levenson-I scale had a Cronbach' alpha of 0.58 ($M(SD)$ = 10.07(5.47), range = 2–24). The Levenson-P subscale had a Cronbach's alpha of 0.71 ($M(SD)$ = 30.11(7.30), range = 8–47). Lastly, the Levenson-C subscale had a Cronbach' alpha of 0.73 ($M(SD)$ = 32.03(7.14), range = 8–47). Scatter plots of all questionnaires by ERN and

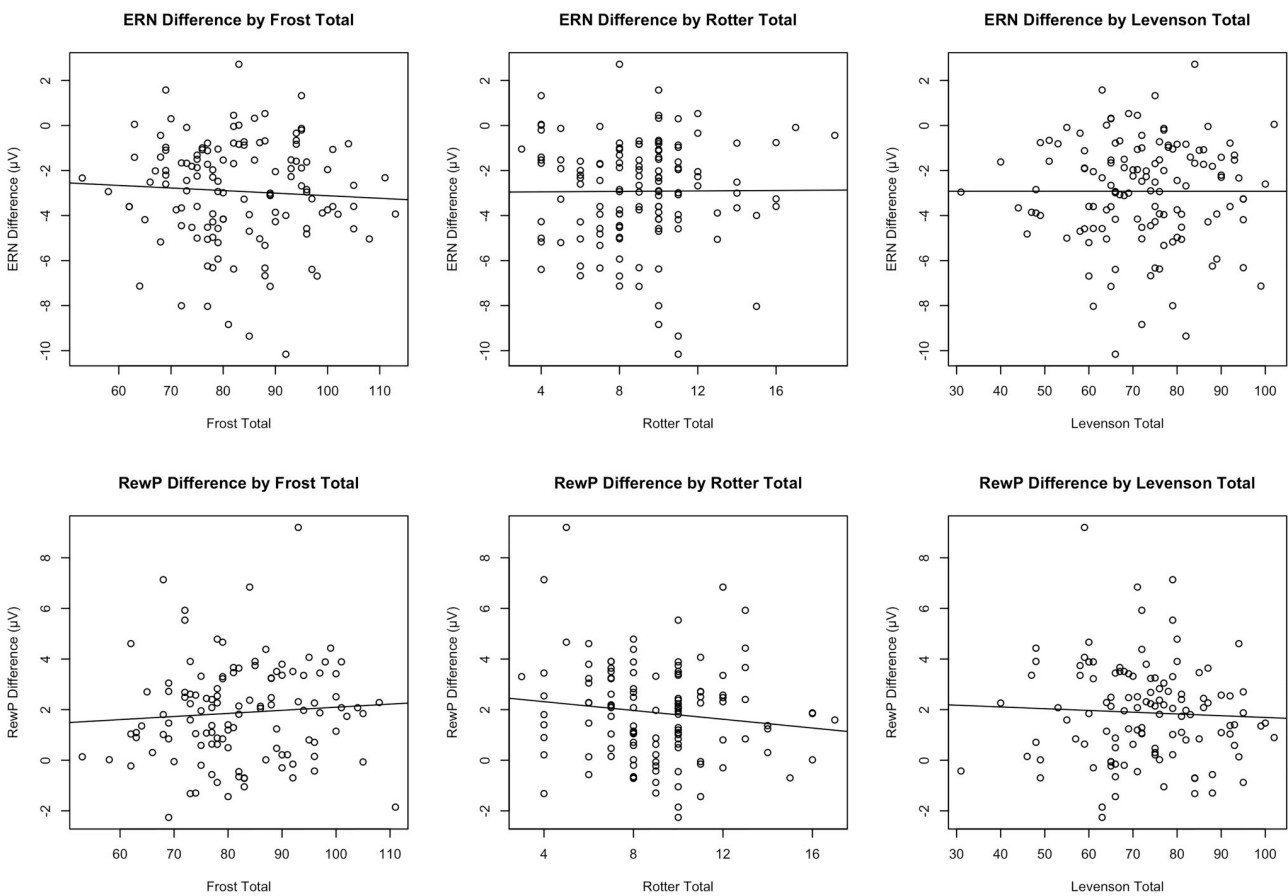

**Fig 1. Scatter plots depicting all scales of interest by ΔERN amplitude or ΔRewP amplitude.**

RewP difference amplitude are shown in Fig 1 signifying adequate range and distribution in questionnaire scores.

After the questionnaires, participants completed two separate computerized tasks in counterbalanced order while electroencephalogram (EEG) data were recorded. First, participants completed a modified version of the Ericksen Flanker task [61]. Incongruent (e.g. $< < > < <$) and congruent (e.g. $< < < < <$) arrow groups were randomly presented in 36-point Arial white font were presented in the center of a black screen. Participants were instructed to respond as quickly and accurately as possible by pressing a button that corresponded to the direction of the middle arrow. Flanking arrows were presented for 100 ms prior to onset of the middle arrow, which remained on the screen with the middle arrow for an additional 600 ms. In between trials, a fixation cross was shown for randomized intervals of 300, 500, and 700 ms. Two blocks of three hundred trials each (600 total trials) were completed with 50% of trials being congruent and 50% of trials being incongruent.

For the doors task [17,30] participants were shown two doors side by side on a black background and were instructed to click the corresponding mouse button to choose a door on either the left or right. Participants were told that if they chose correctly, they would see a green arrow pointing upward, but if they chose incorrectly they would see a red arrow pointing downward. For every correct choice, they would gain 80 cents while they would lose 40 centers for every incorrect choice. Doors were presented until the participant clicked a left or right mouse button; there was no time limit for making the choice. After a door was chosen,

participants were presented with feedback for 2000 ms, although this feedback had no relation to the actual door chosen. Each participant completed 50 trials with 25 wins and 25 losses, for a total of $10. The order of positive or negative feedback was randomized.

## Electroencephalogram Recording and Reduction

EEG data were recorded from 128 equidistant passive Ag/AgCl electrodes on a hydrocel geodesic sensor net from Electrical Geodesics, Inc. using the NA 300 amplifier system (EGI; Eugene, OR; 20K nominal gain, bandpass = .10–100 Hz). Data were referenced to the vertex electrode (Cz) during data collection and digitized continuously at 250 Hz with a 16-bit analog to digital converter. According to manufacturer's instruction, impedances were kept below 50kΩ. Offline, data were digitally high-pass filtered with a first-order 0.1 Hz filter, and digitally low pass filtered at 30 Hz (12 db/octave butterworth filter) in NetStation (version 4.5.7). For the ERN and Pe, data were then segmented from 400 ms prior- to 600 ms post-response for correct and incorrect trials. For the RewP, data were segmented from 200 ms before feedback presentation to 800 ms after feedback. Eye movements and blink artifacts were then corrected using independent components analysis (ICA) in the ERP PCA Toolkit in MatLab [62]. If any ICA component correlated with two blink templates (one template being provided by the ERP PCA Toolkit and one template being derived from previous data by the authors) at a rate of 0.9 or higher, that component was removed from the data. Further, if the fast average amplitude of a particular channel was greater than 100 microvolts or if the differential average amplitude was greater than 50 microvolts, the channel was defined as bad and the nearest neighbor approach (using six electrodes) was used to interpolate the data for said bad electrode [62].

Finally, data were re-referenced offline in the ERP PCA Toolkit using an averaged reference and baseline adjusted from 400 to 200 ms before response for the ERN and Pe and from 200 to 0 ms before the presentation of feedback for the RewP, after which trials were averaged together. The mean amplitude was extracted between 0 and 100 ms for the ERN, between 200 and 400 ms for the Pe, and between 250 and 325 ms for the RewP. The use of a mean amplitude was decided *a priori* due to research suggesting mean amplitude is more reliable than other ERP peak extractions [8,63]. The *a priori* time windows for all three ERPs were decided on through the use of the collapsed localizers approach. The collapsed localizer approach entails collapsing across all groups and variables to view one grand-averaged waveform in order to decide what window to pull mean amplitude from [64]. In order to improve reliability of ERP measurement, we used a region of interest (ROI) for selecting electrodes [65]. For both the ERN and RewP, ERP data were averaged across four fronto-central electrodes (6 (FCz), 7, 106, 129 (Cz); see [66] for electrode montage), as decided *a priori*. Electrode locations were chosen due to previous research suggesting that the ERN and RewP are maximal at these frontocentral locations (e.g., [67]). For the Pe, data were averaged across electrodes 54, 55, 61, 62, 78, and 79, as also decided *a priori*. All ERP component mean amplitudes for all trial types are reported in Table 1.

In order to determine minimum number of trials needed to ensure adequate reliability, dependability estimates of ERP data were assessed through the ERP Reliability Analysis Toolbox v0.3.2 [68]. Dependability estimates for all components are quite high (above 0.83) and are presented in Table 2. For the ERN, a minimum number of 94 correct responses and 6 incorrect responses were required; therefore, 31 participants were excluded from ERN and Pe analyses due to fewer than aforementioned trial numbers. For the RewP, a minimum number of 12 correct feedback trials and 12 incorrect feedback trials were needed. Therefore, 31 participants were excluded from RewP analysis due to lack of sufficient trials. Overall, dependability estimates suggest a high level of reliability, allowing reasonable conclusions to be drawn from

**Table 1. Means and standard deviations for ERP components, task accuracy, and response time.**

|  | Mean | Standard Deviation | Range (min,max) |
|---|---|---|---|
| CRN amplitude (μV) | 1.9 | 1.6 | (-2.9, 5.9) |
| ERN amplitude (μV) | -1.1 | 2.3 | (-7.7, 4.2) |
| ERN difference amplitude (μV) | -2.9 | 2.3 | (-10.2, 2.7) |
| RewP positive feedback (μV) | 5.2 | 3.3 | (-0.01, 20.0) |
| RewP negative feedback (μV) | 3.3 | 2.8 | (-3.3, 15.3) |
| RewP difference amplitude (μV) | 1.9 | 1.9 | (-2.3, 9.2) |
| Pe correct amplitude (μV) | -0.6 | 0.9 | (-3.3, 1.1) |
| Pe incorrect amplitude (μV) | 3.9 | 2.6 | (-1.6, 15.4) |
| Pe difference amplitude (μV) | 4.6 | 2.8 | (-2.4, 18.5) |
| Congruent trial flanker accuracy (%) | 96.5% | 4.2% | (62%, 100%) |
| Incongruent trial flanker accuracy (%) | 90.2% | 7.3% | (59.1%, 99.3%) |
| Post Correct accuracy (%) | 94.2% | 3.8% | (81.6%, 98.6%) |
| Post Error accuracy (%) | 84.0% | 13.5% | (2.68%, 100%) |
| Correct Congruent Flanker RT | 387.8 | 38.7 | (298, 488.5) |
| Correct Incongruent Flanker RT | 459.5 | 35.4 | (382, 555) |
| Incorrect Congruent Flanker RT | 351.5 | 163.8 | (0, 761) |
| Incorrect Incongruent Flanker RT | 302 | 73.8 | (0, 509) |
| Overall Doors RT | 524.5 | 504.1 | (0, 4232) |

*Note*: μV = microvolts. ERN difference amplitude = incorrect minus correct. RewP difference amplitude = correct minus incorrect feedback. Pe difference amplitude = incorrect minus correct.

For all reaction times, the median was calculated.

the data (see Table 2). Due to the non-independence of difference scores, the dependability of difference scores was not calculated. However, exploratory analyses using the residualized difference instead of a subtraction difference are provided below [57].

## Data analysis

**Behavioral data analyses.** Median response times (RT) and mean accuracy are presented for the flanker task as a function of congruency and accuracy and median RT from the doors task (see Table 1). We chose *a priori* to correlate incongruent-trial accuracy and correct-trial incongruent RTs from the flanker task and RT from the doors task with each of the five perfectionism/locus of control scales administered (Frost, Rotter, Levenson I, Levenson P, and Levenson C) to assess if perfectionism or locus of control correlated with behavioral performance during the more cognitively demanding task trials. As a manipulation check, two

**Table 2. ERP dependability and noise estimates.**

| Trial Type | Dependability | 95% Credible Intervals | Minimum Trials | Mean(SD) Trials | Trial Range | Noise Mean(SD) |
|---|---|---|---|---|---|---|
| Correct Response (ERN) | 0.98 | (0.98, 0.99) | 94 | 487.6(83.9) | 94–583 | 0.4(0.4) |
| Incorrect Response (ERN) | 0.83 | (0.78, 0.87) | 6 | 27.8(26.1) | 6–223 | 1.8(1.6) |
| Correct Feedback (RewP) | 0.9 | (0.87, 0.92) | 12 | 1.7(0.4) | 12–25 | 1.7(0.4) |
| Incorrect Feedback (RewP) | 0.87 | (0.83, 0.90) | 12 | 1.8(0.5) | 9–25 | 1.8(0.5) |
| Correct Response (Pe) | 0.95 | (0.94, 0.96) | 94 | 488.8(83.9) | 94–583 | 0.4(0.4) |
| Incorrect Response (Pe) | 0.83 | (0.78, 0.87) | 6 | 27.8(26.1) | 6–223 | 1.8(1.6) |

paired samples *t*-tests comparing accuracy between congruent and incongruent trials and response times between congruent and incongruent trials were conducted for the flanker task.

In order to calculate post-error slowing (the amount a participant's response time slows after an erroneous response [69]), we extracted the RT for every correct trial that was preceded by an error (i.e., post-error RT) and for every correct trial that was followed by an error (i.e., pre-error RT). Pre-error RT was then subtracted from post-error RT to get one value of post-error slowing (for methodology [69]). This was also done for correct trials that were preceded or followed by a correct trial (i.e., pre-correct RT subtracted from post-correct RT; see Table 1). A 2-Accuracy (error slowing, correct slowing) x 2-Trial-type (congruent, incongruent) repeated measures ANOVA was then performed to determine if post error or correct RT slowing was significantly different by trial congruency, with general eta squared used as a measure of effect size. Paired-samples *t*-tests were performed to determine if mean post-error RT differed from mean post-correct trial RT broken apart by congruency with Cohen's $d_z$ used as a measure of effect size. Correlations of error slowing were conducted with the Frost, Rotter, Levenson I, Levenson P, and Levenson C scales.

**ERP analyses.** Three paired samples t-tests were conducted to ensure that ERP effects were present (i.e., ERN amplitude was different than CRN amplitude) for the ERN, RewP, and Pe. In order to test our first hypothesis that individuals with increased perfectionistic tendencies would have greater ERN (more negative) amplitude and smaller (less negative) RewP amplitude compared to those with lower perfectionistic tendencies, we conducted two multiple linear regressions with age, gender (male = 0; female = 1), and total score on the F-MPS predicting ΔERN and ΔRewP amplitude. A third multiple linear regression was conducted with age, gender, and total score on the F-MPS predicting ΔPe.

To test our second hypothesis that individuals with a more internal locus of control would exhibit larger ΔERN amplitudes and smaller ΔRewP amplitudes compared to those with a more external locus of control, we performed eight multiple linear regressions. For the first two regressions, age, gender, and total score on the Rotter scale predicted either ΔERN or ΔRewP. The last six multiple linear regressions had age, gender, and one of the Levenson subscales (Levenson-I, Levenson-C, Levenson-P) predicting either ΔERN or ΔRewP. The subscales were entered into separate regressions, as there is not a total score for the Levenson scale and we wanted to ensure multicollinearity assumptions were met. Four more linear regressions were performed on ΔPe amplitudes with age, gender, and Rotter scale or each of the Levenson subscales predicting difference score amplitude as exploratory analyses.

For all regression models, standardized betas are reported. Adjusted $R^2$, $\Delta R^2$, and Cohen's $f^2$ are reported as measures of effect sizes while variance inflation factor (VIF) scores are reported as measures of multi-collinearity. All models were acceptable for homoscedasticity and met basic assumptions for multicollinearity. Normality of residuals was adequate.

We decided *a priori* that if the models predicting ΔERN difference amplitude were significant, exploratory analysis would be conducted to see whether it was the correct responses (represented by the correct response negativity [CRN]) or the erroneous responses represented by the ERN) that drove significant findings. We also decided *a priori* that if the models including the F-MPS were significant, further exploratory analyses would be completed to see which of the six subscales were significant, but only if the initial analyses were significant.

**Sensitivity analysis and exploratory analyses.** We conducted a sensitivity analysis in G*Power (v3.1) for both the ΔERN and ΔRewP in order to determine what size of an effect we were powered to detect. A linear multiple regression fixed model with $R^2$ deviation from zero was computed for 80% power. For the ΔERN and ΔPe with a final study size of 128, we were powered at 80% to detect an effect size ($f^2$) of at least 0.09, which is between a small and

medium-sized effect. For the ΔRewP with a final sample size of 119, we were powered at 80% to similarly detect an effect size of 0.09.

Due to evidence that difference waves may be insufficiently reliable [55–57], exploratory analyses further investigating the relationship between ERP amplitudes and measures of perfectionism and locus of control were performed in order to ensure that the current results are not due to unreliable data. As an alternative to the difference wave, residuals between the ERP of interest and the opposite ERP (e.g., the error and correct trial waveforms) can be examined [57]. Therefore, twelve additional exploratory linear regression were performed. For the first three regressions, age, gender, and scale score (F-MPS, Rotter, Lev-I, Lev-P, Lev-C) predicted the residuals between the ERN and correct-related negativity (CRN). For the next three regressions, age, gender, and scale score predicted the residuals between the RewP and amplitude values on incorrect feedback trials. For the last three regressions, age, gender, and scale score predicted the residuals between Pe amplitude on correct and error trials. Additionally, twelve linear regressions were performed predicting single ERP amplitude values. The first three regressions used age, gender, and scale score to predict ERN amplitude. The next three regressions used age, gender, and scale score to predict the RewP amplitude. The last three regressions used age, gender, and scale score to predict Pe amplitude.

## Results

### Behavioral data

For the flanker task, paired samples $t$-tests showed that there was greater accuracy for congruent versus incongruent trials ($t(127) = 11.41$, $p < 0.001$, $d_z = 1.07$) and correct-trial RTs were faster for congruent trials than for incongruent trials ($t(127) = 40.61$ $p < 0.001$, $d_z = 1.98$). None of the perfectionism or locus of control scales were significantly correlated with incongruent accuracy on the flanker task, incongruent correct RTs on the flanker task, or overall RTs for the doors task (see S2 Table on OSF for correlation values and $p$-values).

Participants got 94% of trials correct following a correct response ($SD_{post\text{-}correct} = 3.77$), while they answered correctly on only 84% of trials following an erroneous response ($SD_{post\text{-}error} = 13.49$). Accurate post-error trials had a significantly longer RTs than accurate post-correct trials ($t(127) = 12.38$, $p < 0.001$, $d_z = 0.89$), indicative of significant post-error slowing. The 2-Accuracy (error slowing, correct slowing) x 2-Trial-type (congruent, incongruent) ANOVA revealed a main effect of both accuracy ($F_{correct}[1,127] = 216.76$, $p_{correct} < 0.001$, $\eta^2_{correct} = 0.34$) and congruency ($F_{congruency}[1,127] = 18.41$, $p_{congruency} < 0.001$, $\eta^2_{congruency} = 0.03$) with a significant interaction of the two ($F[1,127] = 21.50$, $p < 0.001$, $\eta^2 = 0.04$). Post-hoc paired samples $t$-tests revealed that on both congruent and incongruent trials, participants slowed down significantly more after an error than after a correct response ($t(127) = 14.41$, $p < 0.001$, d = 0.52, $t(127) = 7.95$, $p < 0.001$, $d_z = 1.01$, respectively). No correlations between post-error slowing and the perfectionism or locus of control scales were significant (all r's < .02, ns > .05, see S2 Table on OSF).

### ERP results

See Fig 2 for CRN, ERN, and ΔERN amplitude. See Fig 3 for incorrect feedback, RewP and ΔRewP amplitude. See Fig 4 for incorrect response, correct response, Pe amplitude. All ERP effects were present, namely, ERN amplitude was more negative than CRN amplitude ($t(127) = 14.45$, $p < 0.001$, d = 1.28), Pe error amplitude was more positive than Pe correct-trial amplitude ($t(127) = -18.29$, $p < 0.001$, $d = -1.62$) and RewP amplitude was more positive following reward than non-reward feedback ($t(127) = 10.53$, $p < 0.001$, $d = 0.97$).

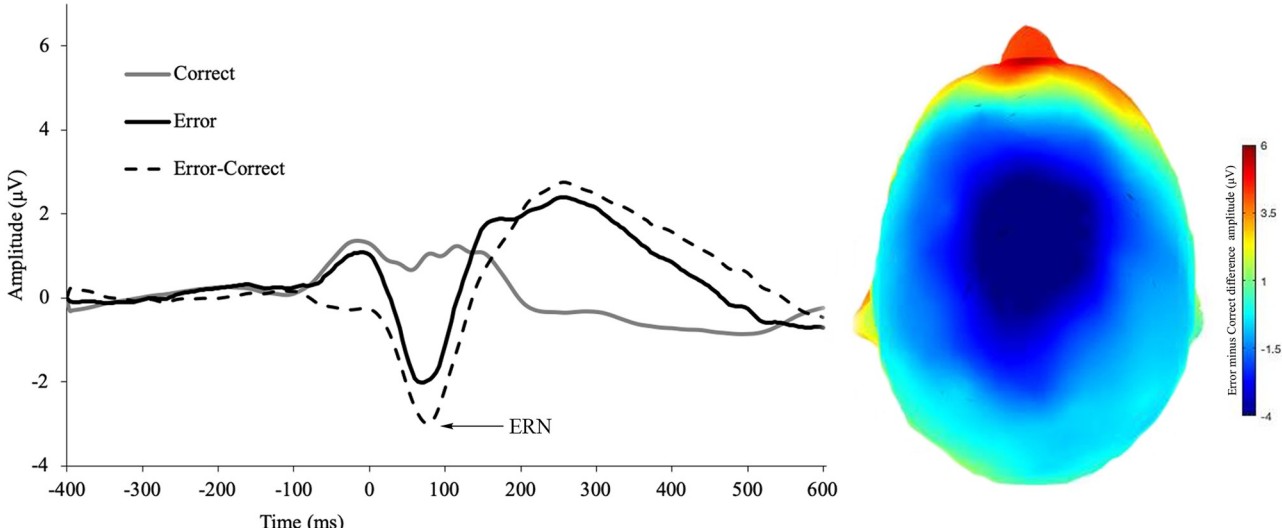

**Fig 2. ERN for erroneous responses, correct responses, and the difference wave during the flanker task.** Scalp distribution of the difference wave (incorrect minus correct responses).

**Frost multi-perfectionism scale.**   Linear regression results for both the perfectionism scale (as measured by the F-MPS as reported below) and locus of control (as measured by the Rotter) are reported in Table 3. When testing our first hypothesis that larger ΔERN but blunted ΔRewP amplitudes would be associated with perfectionistic tendencies as measured by the F-MPS, after controlling for age and gender, F-MPS total scores did not significantly predict ΔERN amplitude ($\beta$ = -0.05, $p$ = 0.55). Similarly, after adjusting for age and gender, F-MPS scores did not predict ΔRewP amplitude ($\beta$ = 0.07, $p$ = 0.44). For the Pe, F-MPS total scores did not predict ΔPe amplitudes ($\beta$ = -0.05, $p$ = 0.58).

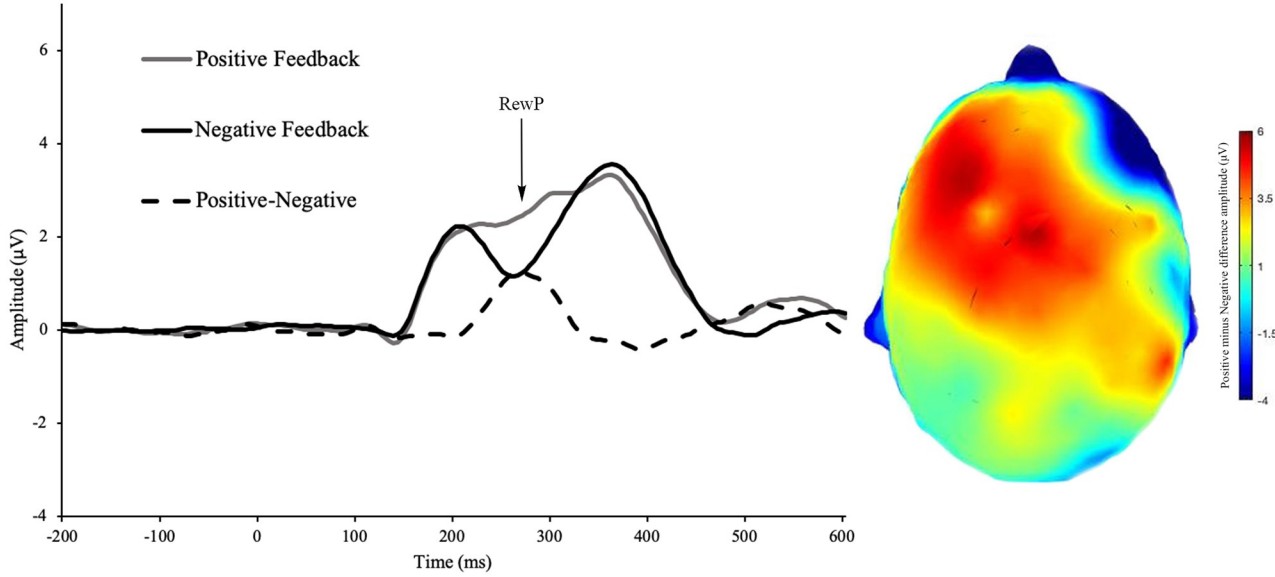

**Fig 3. RewP for correct feedback, incorrect feedback, and the difference wave during the doors task.** Scalp distribution of the difference wave (correct minus incorrect feedback).

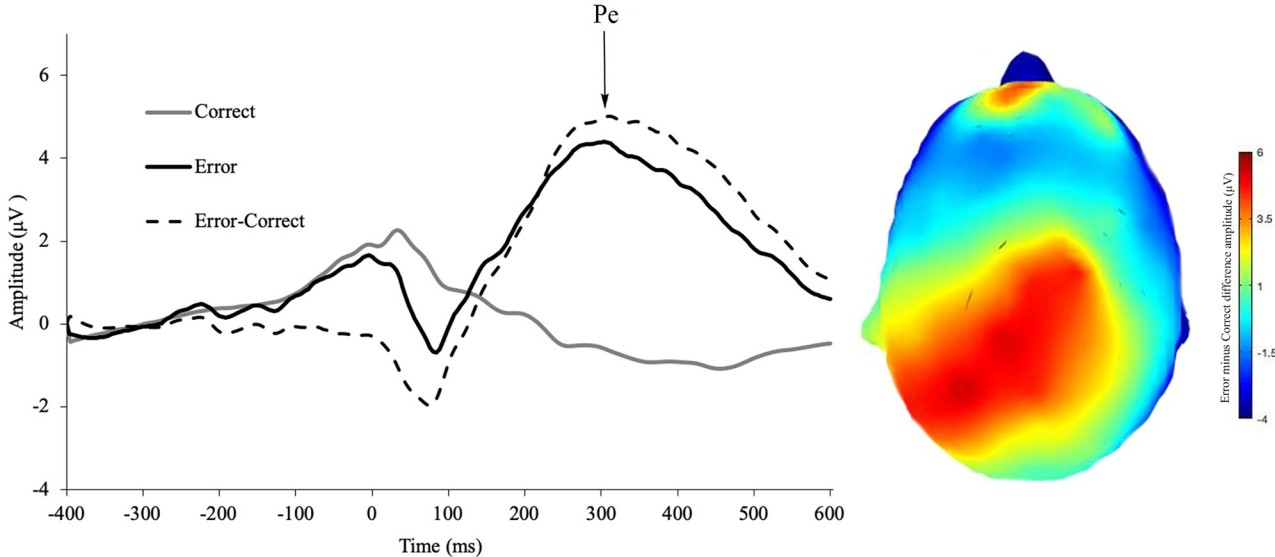

**Fig 4. Pe for erroneous responses, correct responses, and the difference wave during the flanker task.** Scalp distribution of the difference wave (incorrect minus correct responses).

**Rotter scale.** See Table 3 for the results of the linear regressions containing the Rotter scale. When testing our second hypothesis that larger ΔERN and smaller ΔRewP amplitudes would be observed in individuals with a more external locus of control, Rotter total scores ($\beta$ = -0.01, $p$ = 0.88) did not significantly predict ΔERN amplitude. Rotter total scores ($\beta$ = -1.38, $p$ = 0.17) did not significantly predict ΔRewP amplitude. Further, Rotter total score did not predict ΔPe amplitudes ($\beta$ = -0.09, $p$ = 0.33).

**Levenson subscales.** All results for the three Levenson subscales are reported in Table 4. Linear regressions were performed for each Levenson subscale. Similar to the Rotter results, the Levenson-I subscale ($\beta$ = 0.01, $p$ = 0.94) did not predict ΔERN amplitude. Again, the Levenson-P subscale ($\beta$ = -0.01, $p$ = 0.91) did not predict ΔERN amplitude. Finally, the Levenson-C subscale ($\beta$ = 0.03, $p$ = 0.77) did not predict ΔERN amplitude. As a note, in the F-MPS, Rotter, Lev-I, Lev-P, and Lev-C regressions, age did predict ΔERN amplitude when gender and the relevant subscale were adjusted for ($B_{F-MPS}$ = -2.54, $p_{F-MPS}$ = 0.01; $B_{Rotter}$ = -2.57, $p_{Rotter}$ = 0.01; $B_{Lev-I}$ = -2.55, $p_{Lev-I}$ = 0.01; $B_{Lev-P}$ = -2.55, $p_{Lev-P}$ = 0.01; $B_{Lev-C}$ = -2.58, $p_{Lev-C}$ = 0.01). For the ΔRewP multiple linear regressions, none of the Levenson subscales predicted ΔRewP amplitudes ($B_{Lev-I}$ = -0.02, $p_{Lev-I}$ = 0.86; $B_{Lev-P}$ = -0.06, $p_{Lev-P}$ = -0.68; $B_{Lev-C}$ = -0.01, $p_{Lev-C}$ = 0.90). Similarly, none of the Levenson subscales predicted ΔPe amplitudes ($B_{Lev-I}$ = -0.04, $p_{Lev-I}$ = 0.68; $B_{Lev-P}$ = 0.11, $p_{Lev-P}$ = 0.21; $B_{Lev-C}$ = 0.13, $p_{Lev-C}$ = 0.14).

The results of the residual exploratory analyses can be found in Tables 5 and 6. All results of the exploratory analyses matched the previously reported results and showed no statistically-significant predictions between residualized ERN or RewP and perfectionism or locus of control scales.

At the request of a reviewer, additional exploratory linear regressions were conducted with age, sex, and F-MPS, Rotter scale, or Levenson subscale score predicting ERN and RewP amplitude, rather than a difference score or residualized values. The pattern of significance in results from these analyses was the same as those for the difference values presented above. All statistics for the ERN-specific regressions are presented in the supplementary materials available on OSF (S3 Table and S4 Table on OSF).

**Table 3. Multiple linear regressions with Frost Perfectionism scale and Rotter locus of control predicting difference amplitudes.**

| | $\beta$ | $t$ | $\Delta R^2$ | VIF | $F$ | $df$ | Adj. $R^2$ | Cohen's $f^2$ |
|---|---|---|---|---|---|---|---|---|
| **ERN Difference Amplitude Model with Frost** | | | | | 2.5 | 3, 124 | 0.03 | 0.06 |
| Gender | -0.1 | -0.8 | 0.004 | 1.00 | | | | |
| Age | -0.2 | -2.5* | 0.049 | 1.00 | | | | |
| Frost Total | -0.1 | -0.6 | 0.003 | 1.00 | | | | |
| **RewP Difference Amplitude Model with Frost** | | | | | 0.27 | 3, 113 | -0.02 | 0.01 |
| Gender | -0.04 | -0.40 | 0.00 | 1.01 | | | | |
| Age | 0.00 | 0.02 | 0.00 | 1.01 | | | | |
| Frost Total | 0.07 | 0.77 | 0.00 | 1.02 | | | | |
| **Pe Difference Amplitude Model with Frost** | | | | | 1.5 | 3, 124 | 0.01 | 0.04 |
| Gender | -0.1 | -1.5 | 0.02 | 1.00 | | | | |
| Age | -0.1 | -1.3 | 0.02 | 1.00 | | | | |
| Frost Total | -0.1 | -0.6 | 0.002 | 1.00 | | | | |
| **ERN Difference Amplitude Model with Rotter** | | | | | 2.33 | 3, 124 | -0.01 | 0.06 |
| Gender | -0.06 | -0.73 | 0.004 | 1.00 | | | | |
| Age | -0.23 | -2.57* | 0.05 | 1.01 | | | | |
| Rotter Total | -0.01 | -0.15 | < 0.001 | 1.01 | | | | |
| **RewP Difference Amplitude Model with Rotter** | | | | | 0.71 | 3, 113 | -0.01 | 0.02 |
| Gender | -0.04 | -0.48 | 0.00 | 1.01 | | | | |
| Age | -0.01 | -0.05 | 0.00 | 1.01 | | | | |
| Rotter Total | -0.13 | -1.38 | 0.02 | 1.01 | | | | |
| **Pe Difference Amplitude Model with Rotter** | | | | | 1.68 | 3, 124 | 0.02 | 0.04 |
| Gender | -0.13 | -1.51 | 0.02 | 1.00 | | | | |
| Age | -0.13 | -1.49 | 0.02 | 1.01 | | | | |
| Rotter Total | -0.09 | -0.98 | 0.01 | 1.01 | | | | |

*Note.* VIF = variance inflation factor.

*$p$<.05.

**$p$<.01.

***$p$<.001.

## Discussion

Our central purpose was to examine the relationship between perfectionism, locus of control, and the ΔERN and ΔRewP ERP components in a psychiatrically-healthy sample to see if specific personality traits are related to internal and external performance monitoring. Our first hypothesis that individuals with increased perfectionistic tendencies would have greater ΔERN amplitude and smaller ΔRewP amplitude compared to those with lower perfectionistic tendencies was not supported as we found that perfectionistic traits were not related to indices of internal nor external performance monitoring. Further, our second hypothesis that individuals with a more internal locus of control would exhibit larger ΔERN amplitudes and smaller ΔRewP amplitudes when compared to those with a more external locus of control was also not supposed as we found that locus of control, whether internal or external, did not associate with either internal or external performance monitoring. Similarly, the behavioral outcomes (i.e., response times, post-error slowing, and accuracy) were not related to perfectionism or locus of control personality traits.

The current body of literature concerning perfectionism and ERN amplitude suggests that perfectionism may not be related to ERN amplitude, although specific subscales of the F-MPS may be. As with the current study, Schrijvers et al. (2010) found no significant impact of total

**Table 4. Multiple linear regressions with Levenson subscales (locus of control) predicting difference amplitudes.**

| | β | t | ΔR² | VIF | F | df | Adj. R² | Cohen's f² |
|---|---|---|---|---|---|---|---|---|
| **ERN Difference Amplitude Model with Lev I** | | | | | 2.32 | 3, 124 | 0.03 | 0.05 |
| Gender | -0.07 | -0.73 | 0.00 | 1.02 | | | | |
| Age | -0.22 | -2.55* | 0.05 | 1.01 | | | | |
| Lev I Total | 0.01 | 0.08 | 0.00 | 1.03 | | | | |
| **RewP Difference Amplitude Model with Lev I** | | | | | 0.08 | 3, 113 | -0.02 | 0.002 |
| Gender | -0.04 | -0.43 | 0.00 | 1.02 | | | | |
| Age | 0.01 | 0.08 | 0.00 | 1.01 | | | | |
| Lev I Total | -0.02 | -0.18 | 0.00 | 1.02 | | | | |
| **Pe Difference Amplitude Model with Lev I** | | | | | 1.41 | 3, 124 | 0.01 | 0.03 |
| Gender | -0.13 | -1.42 | 0.02 | 1.03 | | | | |
| Age | -0.13 | -1.43 | 0.02 | 1.01 | | | | |
| Lev I Total | -0.04 | -0.42 | 0.00 | 1.03 | | | | |
| **ERN Difference Amplitude Model with Lev P** | | | | | 2.33 | 3, 124 | 0.03 | 0.06 |
| Gender | -0.06 | -0.73 | 0.00 | 1.01 | | | | |
| Age | -0.22 | -2.55* | 0.05 | 1.01 | | | | |
| Lev P Total | -0.01 | -0.11 | 0.00 | 1.01 | | | | |
| **RewP Difference Amplitude Model with Lev P** | | | | | 0.22 | 3, 113 | -0.02 | 0.001 |
| Gender | -0.05 | -0.49 | 0.00 | 1.00 | | | | |
| Age | 0.01 | 0.12 | 0.00 | 1.00 | | | | |
| Lev P Total | -0.06 | -0.68 | 0.00 | 1.00 | | | | |
| **Pe Difference Amplitude Model with Lev P** | | | | | 1.9 | 3, 124 | 0.02 | 0.05 |
| Gender | -0.12 | -1.37 | 0.02 | 1.01 | | | | |
| Age | -0.13 | -1.5 | 0.02 | 1.01 | | | | |
| Lev P Total | 0.11 | 1.27 | 0.01 | 1.02 | | | | |
| **ERN Difference Amplitude Mode with Lev C** | | | | | 2.35 | 3, 124 | 0.03 | 0.06 |
| Gender | -0.06 | -0.74 | 0.00 | 1.00 | | | | |
| Age | -0.23 | -2.58* | 0.05 | 1.01 | | | | |
| Lev C Total | 0.03 | 0.3 | 0.00 | 1.01 | | | | |
| **RewP Difference Amplitude Model with Lev C** | | | | | 0.08 | 3, 113 | -0.03 | 0.002 |
| Gender | -0.04 | -0.45 | 0.00 | 1.01 | | | | |
| Age | 0.01 | 0.11 | 0.00 | 1.01 | | | | |
| Lev C Total | -0.01 | -0.13 | 0.00 | 1.02 | | | | |
| **Pe Difference Amplitude Model with Lev C** | | | | | 2.12 | 3, 124 | 0.03 | 0.05 |
| Gender | -0.14 | -1.56 | 0.02 | 1.01 | | | | |
| Age | -0.13 | -1.52 | 0.02 | 1.01 | | | | |
| Lev C Total | 0.13 | 1.5 | 0.02 | 1.01 | | | | |

*Note*: VIF = variance of inflation factor.

*$p<.05$.

**$p<.01$.

***$p<.001$.

F-MPS scores on ΔERN amplitude in a depressed sample. However, although total F-MPS score may not be related to ERN amplitude, numerous studies have suggested that certain subscales of perfectionism, such as personal standards, concern over mistakes, and doubts about actions, may affect ERN amplitude [45,46]. Stahl et al. [45], when investigating the ERN, suggests that it may be the interaction of these subscales, such as high personal standards and

**Table 5. Multiple linear regressions with Frost Perfectionism scale and Rotter locus of control scale predicting residual values.**

| | $\beta$ | $t$ | $\Delta R^2$ | VIF | $F$ | df | Adj. $R^2$ | Cohen's $f^2$ |
|---|---|---|---|---|---|---|---|---|
| **ERN Residual Model with Frost** | | | | | 2.75 | 3,124 | 0.04 | 0.07 |
| Gender | -0.11 | -1.25 | .004 | 1.00 | | | | |
| Age | -0.22 | -2.52* | 0.04 | 1.00 | | | | |
| Frost Total | -0.06 | -0.68 | < 0.001 | 1.00 | | | | |
| **RewP Residual Model with Frost** | | | | | 0.26 | 3,113 | -0.02 | 0.01 |
| Gender | -0.03 | -0.30 | 0.00 | 1.01 | | | | |
| Age | -0.01 | -0.12 | 0.00 | 1.01 | | | | |
| Frost Total | 0.08 | 0.81 | 0.01 | 1.02 | | | | |
| **Pe Residual Model with Frost** | | | | | 1.07 | 3,124 | 0.00 | 0.03 |
| Gender | -0.14 | -1.6 | 0.02 | 1.00 | | | | |
| Age | -0.06 | -0.68 | -0.001 | 1.00 | | | | |
| Frost Total | -0.04 | -0.54 | -0.006 | 1.00 | | | | |
| **ERN Residual Model with Rotter** | | | | | 2.58 | 3,124 | 0.04 | 0.06 |
| Gender | -0.11 | -1.23 | 0.004 | 1.00 | | | | |
| Age | -0.22 | -2.54* | 0.04 | 1.01 | | | | |
| Rotter Total | 0 | -0.03 | -0.008 | 1.01 | | | | |
| **RewP Residual Model with Rotter** | | | | | 0.68 | 3,113 | -0.01 | 0.02 |
| Gender | -0.04 | -0.38 | 0.00 | 1.00 | | | | |
| Age | -0.02 | -0.19 | 0.00 | 1.01 | | | | |
| Rotter Total | -0.13 | -1.38 | 0.02 | 1.01 | | | | |
| **Pe Residual Model with Rotter** | | | | | 1.19 | 3,124 | 0.01 | 0.03 |
| Gender | -0.14 | -1.59 | 0.01 | 1.00 | | | | |
| Age | -0.07 | -0.77 | -0.003 | 1.01 | | | | |
| Rotter Total | -0.07 | -0.82 | -0.003 | 1.01 | | | | |

*Note.* VIF = variance inflation factor.

*$p<.05$.

**$p<.01$.

***$p<.001$.

concern over mistakes, that moderate ERN amplitude in individuals. Although the previously cited studies support the current findings of no to a small relationship between perfectionism and ERN amplitude, Pieters et al. [70] demonstrated a significant correlation between ΔERN amplitude and F-MPS total scores, but only in controls and not in individuals with anorexia nervosa, who had a higher average score of perfectionism. Overall, it seems that total F-MPS scores is not likely related to ERN amplitude.

The current results also suggest that there is no relationship between locus of control (neither external nor internal) and performance monitoring. It is possible that locus of control depends on the situation at hand, rather than being a stable personality trait. Rotter (1975) suggested that classifying people as having strictly an internal locus of control or external locus of control does not capture the entirety of the concept of locus of control. For example, individuals may have a more external locus of control in one situation but in other situations exhibit a more internal locus of control [71,72]. This phenomenon is called bilocal expectancy, dual control, or shared responsibility [72]. Bilocal expectancy could make it particularly difficult to parse relationships between performance monitoring ERP components and locus of control due to potential changes in the loci of control.

**Table 6. Multiple linear regressions with Levenson subscales locus of control predicting residual values.**

| | $\beta$ | $t$ | $\Delta R^2$ | VIF | $F$ | $df$ | Adj. $R^2$ | Cohen's $f^2$ |
|---|---|---|---|---|---|---|---|---|
| **ERN Residual Value Model with Lev I** | | | | | 2.65 | 3,124 | 0.04 | 0.06 |
| Gender | -0.11 | -1.28 | 0.005 | 1.03 | | | | |
| Age | -0.22 | -2.51* | 0.04 | 1.01 | | | | |
| Lev I Total | 0.04 | 0.42 | -0.009 | 1.03 | | | | |
| **RewP Residual Value Model with Lev I** | | | | | 0.06 | 3,113 | -0.03 | 0.00 |
| Gender | -0.03 | -0.32 | 0.00 | 1.02 | | | | |
| Age | -0.00 | -0.06 | 0.00 | 1.01 | | | | |
| Lev I Total | -0.02 | -0.24 | 0.00 | 1.02 | | | | |
| **Pe Residual Model with Lev I** | | | | | 1.01 | 3,124 | 0.00 | 0.02 |
| Gender | -0.14 | -1.51 | 0.01 | 1.02 | | | | |
| Age | -0.07 | -0.73 | -0.004 | 1.01 | | | | |
| Lev I Total | -0.03 | -0.34 | -0.007 | 1.03 | | | | |
| **ERN Residual Value Model with Lev P** | | | | | 2.66 | 3,124 | 0.04 | 0.06 |
| Gender | -0.11 | -1.27 | 0.005 | 1.01 | | | | |
| Age | -0.22 | -2.51* | 0.04 | 1.01 | | | | |
| Lev P Total | -0.04 | -0.45 | -0.006 | 1.02 | | | | |
| **RewP Residual Value Model with Lev P** | | | | | 0.17 | 3,113 | -0.02 | 0.01 |
| Gender | -0.04 | -0.39 | 0.00 | 1.00 | | | | |
| Age | -0.00 | -0.02 | 0.00 | 1.00 | | | | |
| Lev P Total | -0.06 | -0.61 | 0.00 | 1.00 | | | | |
| **Pe Residual Value Model with Lev P** | | | | | 1.11 | 3,124 | 0.00 | 0.03 |
| Gender | -0.14 | -1.51 | 0.01 | 1.01 | | | | |
| Age | -0.07 | -0.75 | -0.004 | 1.01 | | | | |
| Lev P Total | 0.06 | 0.66 | -0.005 | 1.02 | | | | |
| **ERN Residual Value Model with Lev C** | | | | | 2.58 | 3,124 | 0.04 | 0.06 |
| Gender | -0.11 | -1.23 | 0.004 | 1.00 | | | | |
| Age | -0.22 | -2.54* | 0.04 | 1.01 | | | | |
| Lev C Total | 0.00 | 0.02 | -0.008 | 1.01 | | | | |
| **RewP Residual Value Model with Lev C** | | | | | 0.05 | 3,113 | -0.03 | 0.00 |
| Gender | -0.03 | -0.35 | 0.00 | 1.01 | | | | |
| Age | -0.00 | -0.03 | 0.00 | 1.01 | | | | |
| Lev C Total | -0.01 | -0.10 | 0.00 | 1.02 | | | | |
| **Pe Residual Value Model with Lev C** | | | | | 1.27 | 3,124 | 0.01 | 0.03 |
| Gender | -0.14 | -1.62 | 0.01 | 1.00 | | | | |
| Age | -0.07 | -0.77 | -0.003 | 1.01 | | | | |
| Lev C Total | 0.08 | 0.95 | < -0.001 | 1.01 | | | | |

*Note.* VIF = variance inflation factor.

*$p<.05$.

**$p<.01$.

***$p<.001$.

Findings from the current study should be understood in the context of limitations. First, only a psychiatrically-healthy sample of undergraduates with no psychopathology was examined, therefore removing any effects that psychopathology may have on performance monitoring. Examining only a healthy sample was done in order to control for any confounding affects psychopathology may have; however, we recognize that it limits our abilities to interpret these

findings in a psychopathology context and prevents us from understanding how or if these traits would differentially affect the ERN and RewP in a sample with psychopathologies.

Therefore, future research should examine the relationship between perfectionism, locus of control, and performance monitoring ERP components in individuals with psychopathologies that exhibit high levels of perfectionistic traits, such as obsessive-compulsive disorder, obsessive-compulsive personality disorder, eating disorders, and anxiety disorders. Further, it may be useful in future research to look other measures of perfectionism, such as the Hewitt-Flett Multidimensional Perfectionism Scale, which examines other sub-dimensions of perfectionism, such as self-oriented perfectionism, that are not measured in the F-MPS [73]. Finally, although we did run a wide number of analyses, findings do not suggest false positives due to Type I error as the pattern was that of non-significance.

Although there are several limitations, there are also several strengths in our study. After performing a post-hoc sensitivity analysis, the results suggest that our study was well powered to detect a small-to-medium sized effect. Therefore, we feel confident that if a small effect had been present, we would have been able to detect it, and that our final results are less likely due to Type II error. Another strength of our study is that we measured locus of control through two different scales, therefore allowing us to test the possibility that sub-dimensions of locus of control would be related to performance monitoring. Lastly, we had a well-controlled sample that was free of potential confounding variables, such as neurological diseases, learning disabilities, or any head injuries that resulted in unconsciousness. Therefore, we can be fairly confident that in healthy individuals, perfectionism and locus of control are not personality characteristics that affect performance monitoring, as measured by the ERN and RewP.

In conclusion, in the current sample, perfectionism and locus of control were not related to neural indices of internal or external performance monitoring. Future research should examine this in clinical populations or explore other characteristic traits, such as worry, that may affect performance monitoring ERP components. As we come to better understand how internal and external performance monitoring differ, we can better understand what specific cognitive deficits are present in psychopathologies, therefore aiding in diagnoses and treatment.

## Author Contributions

**Conceptualization:** Kaylie A. Carbine, Tanja Endrass, Michael J. Larson.

**Data curation:** Kaylie A. Carbine, Jayden Goodwin, Michael J. Larson.

**Formal analysis:** Alexandra M. Muir, Ariana Hedges-Muncy.

**Funding acquisition:** Kaylie A. Carbine.

**Investigation:** Alexandra M. Muir, Kaylie A. Carbine, Jayden Goodwin, Tanja Endrass.

**Methodology:** Kaylie A. Carbine, Tanja Endrass, Michael J. Larson.

**Project administration:** Kaylie A. Carbine, Jayden Goodwin.

**Software:** Alexandra M. Muir.

**Supervision:** Michael J. Larson.

**Writing – original draft:** Alexandra M. Muir.

**Writing – review & editing:** Alexandra M. Muir, Kaylie A. Carbine, Ariana Hedges-Muncy, Tanja Endrass, Michael J. Larson.

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
