## [Decision Letter · Decision Letter 0]

10 Sep 2019

PONE-D-19-18584

Differentiating electrophysiological indices of internal and external performance monitoring: Relationship with perfectionism and locus of control

PLOS ONE

Dear Dr. Alexandra Michelle Muir,

Thank you for submitting your manuscript to PLOS ONE. After careful consideration, we feel that it has merit but does not fully meet PLOS ONE’s publication criteria as it currently stands. Therefore, we invite you to submit a revised version of the manuscript that addresses the points raised during the review process.

I really enjoyed your article and that I believe it should be accepted for publication in PLOS ONE with minor corrections, after you address the concerns of the reviewers, meanly Reviewer 1.

We would appreciate receiving your revised manuscript by November 8, 2019. To enhance the reproducibility of your results, we recommend that if applicable you deposit your laboratory protocols in protocols.io, where a protocol can be assigned its own identifier (DOI) such that it can be cited independently in the future. For instructions see: http://journals.plos.org/plosone/s/submission-guidelines#loc-laboratory-protocols

We look forward to receiving your revised manuscript.

Kind regards,

Thalia Fernandez, Ph.D.

Academic Editor

PLOS ONE

Journal Requirements:

[A Brigham Young University Mentored Environment Grant and the Brigham Young University

527 College of Family, Home, and Social Sciences funded this research.]

 [The authors received no specific funding for this work.]

3. Please upload a copy of Supporting Information Table S1 which you refer to in your text on page 10.

Additional Editor Comments (if provided):

Dear Dr. Muir,

I am very sorry for having delayed the decision on your manuscript, but there were circumstances that did not allow me to decide before.

Kind regards,

Thalía

Reviewers' comments:

Reviewer's Responses to Questions

**Comments to the Author**

1. Is the manuscript technically sound, and do the data support the conclusions?

Reviewer #1: Yes

Reviewer #2: Yes

2. Has the statistical analysis been performed appropriately and rigorously? 

Reviewer #1: Yes

Reviewer #2: Yes

3. Have the authors made all data underlying the findings in their manuscript fully available?

Reviewer #1: Yes

Reviewer #2: Yes

4. Is the manuscript presented in an intelligible fashion and written in standard English?

Reviewer #1: Yes

Reviewer #2: Yes

5. Review Comments to the Author

Reviewer #1: In this manuscript the authors explored the association between individual difference measures of perfectionism, locus of control, and ERN, Pe, and RewP.

In general, the manuscript is clear and well written. Methods follow the research questions and results allow to answer such questions. First of all, I’d like to congratulate the authors for all the control analyses they carried out in order to reduce the chance of getting false positives. My concerns are mainly about the statistical methodology, and I would be happy to review this paper again after they are answered or fulfilled by the authors.

ABSTRACT

1. Please define Pe.

INTRODUCTION

2. What do the authors mean with individual difference traits? Aren’t they just individual traits?

3. The authors claim that “Another ERP component where individual differences, such as levels of anhedonic depression, are implicated is the reward positivity”. I wonder whether there is any ERP for which there are no individual differences, like age, gender, fatigue, psychopathology, etc. maybe the authors might emphasize that they refer to individual personality or trait differences.

HYPOTHESIS

4. I wonder why would individuals with more internal locus of control exhibit enhanced ERN and reduced RewP? Given that they do not attribute high control over themselves and give most of the credit to the environment, one would expect for them to show enhanced sensitivity to external feedback, and hence larger RewP amplitudes. Please comment.

5. I don’t get why the authors suggest that heightened Pe would be present in participants with increased perfectionism AND those with more internal locus of control. From the Introduction, one would think that perfectionists do not trust their own abilities to influence environment, so then enhanced Pe would be expected in perfectionist participants with low internal locus.

METHODS

6. The authors decided to use only differential waves for their analyses, and they give an argument about that. Certainly, differential waves can extract reliably the components of interest. Nevertheless, I think it is always useful to explore the raw ERP that contains the process of interest, as these data are less manipulated. Did the authors try using these ERP for their regression analyses?

7. While there were no lineal associations between ERP and personality traits, I wonder whether there would be a difference if ERP were compared in terms of presence/absence of trait. Did the authors try comparing the ERP by a median split within the sample or based on cutoff points? Moreover, as they did not include participants with evident psycopathology, how about comparing participants with extreme scores in each scale? Even if the sample did not include psychiatric patients, a high score in the scales (e.g. 48 in the Levenson scale) could be considered as a risk factor.

8. Were ERN and RewP correlated across participants?

DISCUSSION

Please propose some clinical populations in which it would be relevant to explore the association between these personality traits and ERP.

MINOR

Line 259, page 12, “cents” instead of “centers”

Reviewer #2: Experiments seem to have been performed in an adequate manner. A-priori analyses were carried out to ensure an adequate reliability of the ERPs. Achieved power, effect sizes and other important values are reported. Data underlying their findings is readily available. The manuscript is presented in a good English.

6. PLOS authors have the option to publish the peer review history of their article (what does this mean?). If published, this will include your full peer review and any attached files.

Reviewer #1: No

Reviewer #2: Yes: Mauricio González-López

---

## [Author Response · Author response to Decision Letter 0]

13 Sep 2019

We thank both the reviewers and the editors for their comments and thoughts on the previous version of the manuscript. We have addressed each of the comments below and in the manuscript. Changes to the manuscript are highlighted in red font in one version and a clean version is also included. Our specific responses to each comment are included below.

Journal Requirements

1. To our knowledge, all files have now been named and formatted in accordance with PLOS ONE’s guidelines. Please let us know if there are any additional inaccuracies in this regard.

2. We would like to update our Funding Statement to the following: “A Brigham Young University Mentored Environment Grant and the Brigham Young University College of Family, Home, and Social Sciences funded this research.” 

This statement has been removed from the text of the manuscript and is now only present in the funding statement. 

3. Please upload a copy of Supporting Information Table S1 which you refer to in your text on page 10. 

Supporting Materials have been uploaded. The PDF version of each supporting file has been uploaded so that hyperlinks can be provided in the paper. The zipped file named “Supporting Information” contains PDF versions of each table separately so that hyperlinks can be provided in the paper.

4. Please include captions for your Supporting Information files at the end of your manuscript, and update any in-text citations to match accordingly. 

Captions have been added at the end of the manuscript and in-text citations have been double-checked to ensure accuracy. 

Reviewer 1

ABSTRACT

1. Please define Pe.

This change has been made. The Pe is now first referred to as the error-positivity (Pe) in the abstract.

INTRODUCTION

2. What do the authors mean with individual difference traits? Aren’t they just individual traits?

For this manuscript, we specifically are referring to individual differences in personality traits (Sackett et al., 2017). We appreciate the reviewer’s suggestion and understand that the term individual differences encompasses many different facets, including cognition and vocational differences, along with personality traits. Therefore, the manuscript has been updated to elucidate that we are specifically interested in individual differences in personality traits through the following sentence: “A growing consensus indicates that personality traits that vary across individuals, such as anxious apprehension, are also consistently associated with increased neural indices of performance monitoring (7).”

Sackett, Iddekinge, Lievens, & Kuncel (2017). Individual differences and their measurement: A review of 100 years of research. Journal of Applied Psychology. Advance online publication. http://dx.doi.org/10.1037/apl0000151

3. The authors claim that “Another ERP component where individual differences, such as levels of anhedonic depression, are implicated is the reward positivity.” I wonder whether there is an ERP for which there are no individual differences, like age, gender, fatigue, psychopathology, etc. maybe the authors might emphasize that they refer to individual personality or trait differences.

We thank the reviewer for this thoughtful comment and agree that our wording was a bit broad. The manuscript has been updated to say the following: “Another ERP component where personality trait differences, such as levels of anhedonic depression, are implicated is the reward positivity (RewP; (18)).” 

HYPOTHESIS

4. I wonder why would individuals with more internal locus of control exhibit enhanced ERN and reduced RewP? Given that they do not attribute high control over themselves and give most of the credit to the environment, one would expect for them to show enhanced sensitivity to external feedback, and hence larger RewP amplitudes. Please comment.

We believe we have similar hypotheses to what the reviewer suggests. An individual with an internal locus of control would internalize credit more strongly, while someone with external locus of control would tend to externalize the thoughts of power or credit. For example, individuals with a more internal locus of control will attribute outcomes and reinforcements to their own skills, behavior, and abilities (Rotter, 1989; citation 49 in the manuscript); in other words, an individual with internal locus of control will tend to attribute high control over themselves. Conversely, those with an external locus of control will attribute outcomes and reinforcements to factors in the environment, therefore believing that the outcome had less to do with their individual abilities. Thus, we hypothesized that individuals with a more internal locus of control would show greater �ERN amplitude due to greater internal performance monitoring. Conversely, it was hypothesized that we would see a reduced �RewP, again due to high internal, rather than external, monitoring processes.

In order to make this clear in the manuscript, changes have been made on lines 140-142 that read as follows: “For example, an individual with an internal locus of control will attribute outcomes to internal factors, such as skills and abilities. Therefore, errors may be increasingly salient to those who have an internal locus of control when compared to those who attribute errors not to personal reasons, but to external sources, such as the environment.”

5. I don’t get why the authors suggest that heightened Pe would be present in participants with increased perfectionism AND those with a more internal locus of control. From the introduction, one would think that perfectionists do not trust their own abilities to influence environment, so then enhanced Pe would be expected in perfectionistic participants with low internal locus.

The error-positivity, as currently conceptualized, is thought to reflect the conscious awareness of errors (Falkenstein, Christ, Hohnbein, 2000). Those with a high internal locus of control place much more emphasis on controlling behavior, as they believe that they as an individual, due to their unique skills and abilities, can influence performance outcomes. Therefore, we hypothesized that individuals with a high internal locus of control would exhibit a greater �Pe amplitude, as conscious awareness of errors would be higher than those who believe that their actions have less of an influence on their performance. 

In addition, no previous studies have examined the Pe in relation to locus of control, so our hypotheses are exploratory. While arguments can be made either direction, we hypothesized that greater internal locus of control (along with greater levels of perfectionism) would be related to increased Pe amplitude, as internal locus of control is positively correlated with perfectionism dimensions (Jackman, Thorsteinsson, & McNeil, 2017; Citation 50 in the manuscript).

METHODS

6. The authors decided to use only differential waves for their analyses, and they give an argument about that. Certainly, differential waves can extract reliably the components of interest. Nevertheless, I think it is always useful to explore the raw ERP that contains the process of interest, as these data are less manipulated. Did the authors try using these ERP for their regression analyses?

We have now completed regressions using the ERN only (not the difference). Findings from these regressions mirrored the pattern of results presented in the current manuscript. We have added a paragraph to the manuscript on lines 457 to 462 and included a table in the supplemental material with these results. The change to the manuscript reads as follows: “At the request of a reviewer, additional exploratory linear regressions were conducted with age, sex, and F-MPS, Rotter scale, or Levenson subscale score predicting ERN and RewP amplitude, rather than a difference score or residualized values. The pattern of significance in results from these analyses was the same as those for the difference values presented above. All statistics for the ERN-specific regressions are presented in the supplementary materials (Table S3, Table S4).”

7. While there were no lineal associations between ERP and personality traits, I wonder whether there would be a difference if ERP were compared in terms of presence/absence of trait? Did the authors try comparing the ERP by a median split within the sample or based on cutoff points? Moreover, as they did not include participants with evident psychopathology, how about comparing participants with extreme scores in each scale? Even if the sample did not include psychiatric patients, a high score in the scales (e.g. 48 in the Levenson scale) could be considered a risk factor.

We appreciate the reviewer’s comment and thoughts. The goal of the study was to look at continuous relationships between ERP values and personality trait measures. As such, we did not try comparing the ERP by a median split within the sample or based on cutoff points. This decision was made due to the fact that categorizing data based on a median split causes a significant loss in data and statistical power (Cohen, 1983). One of the strengths of the current study is the large sample size, leading to adequate power to detect even a small effect. Therefore, we wanted to utilize the full sample and not put categories onto a continuous variable. If the reviewer feels very strongly this would contribute to the manuscript, we would be willing to add them. However, we would prefer to leave the continuous data as is given the very consistent pattern of the findings and not make artificial categories.

8. Were the ERN and RewP correlated across participants?

a. The ERN and the RewP were not correlated across participants (r(114) = -0.17, p = 0.07).

DISCUSSION

Please propose some clinical populations in which it would be relevant to explore the association between these personality traits and ERP.

The following sentence has been added to the discussion section (lines 508-511): “Therefore, future research should examine the relationship between perfectionism, locus of control, and performance monitoring ERP components in individuals with psychopathology that exhibit high levels of perfectionistic traits, such as obsessive-compulsive disorder, obsessive-compulsive personality disorder, eating disorders, and anxiety disorders.”

Thank you for this suggestion. 

MINOR

1. Line 259, page 12, “cents” instead of “centers”

a. This change has been made. Thank you for catching the mistake.

---

## [Decision Letter · Decision Letter 1]

14 Oct 2019

Differentiating electrophysiological indices of internal and external performance monitoring: Relationship with perfectionism and locus of control

PONE-D-19-18584R1

Dear Dr. Muir,

We are pleased to inform you that your manuscript has been judged scientifically suitable for publication and will be formally accepted for publication once it complies with all outstanding technical requirements.

With kind regards,

Thalia Fernandez, Ph.D.

Academic Editor

PLOS ONE

Additional Editor Comments (optional):

Reviewers' comments:

Reviewer's Responses to Questions

**Comments to the Author**

1. If the authors have adequately addressed your comments raised in a previous round of review and you feel that this manuscript is now acceptable for publication, you may indicate that here to bypass the “Comments to the Author” section, enter your conflict of interest statement in the “Confidential to Editor” section, and submit your "Accept" recommendation.

Reviewer #1: All comments have been addressed

Reviewer #2: All comments have been addressed

2. Is the manuscript technically sound, and do the data support the conclusions?

Reviewer #1: (No Response)

Reviewer #2: Yes

3. Has the statistical analysis been performed appropriately and rigorously? 

Reviewer #1: (No Response)

Reviewer #2: Yes

4. Have the authors made all data underlying the findings in their manuscript fully available?

Reviewer #1: (No Response)

Reviewer #2: Yes

5. Is the manuscript presented in an intelligible fashion and written in standard English?

Reviewer #1: (No Response)

Reviewer #2: Yes

6. Review Comments to the Author

Reviewer #1: (No Response)

Reviewer #2: The comments made by reviewer 1 were adequately addressed, the research question is well presented, as well as the psychophysiological correlates of the psychological constructs posed.

7. PLOS authors have the option to publish the peer review history of their article (what does this mean?). If published, this will include your full peer review and any attached files.

Reviewer #1: No

Reviewer #2: Yes: Mauricio González-López

---

## [Editor Report · Acceptance letter]

23 Oct 2019

PONE-D-19-18584R1 

Differentiating electrophysiological indices of internal and external performance monitoring: Relationship with perfectionism and locus of control 

Dear Dr. Muir:

I am pleased to inform you that your manuscript has been deemed suitable for publication in PLOS ONE. Congratulations! Your manuscript is now with our production department. 

With kind regards,

on behalf of

Dr. Thalia Fernandez 

Academic Editor

PLOS ONE